# A Patient with Erdheim-Chester Disease Limited to Central Nervous System

Rajesh K. Gupta [1,*], Anam Haque [2], Thejasvi A. Reddy [2] and Carlos A. Pérez [3]

1 Division of Multiple Sclerosis and Neuroimmunology, Department of Neurology, University of Texas Health Science Center at Houston, Houston, TX 77030, USA

2 McGovern Medical School, University of Texas Health Science Center at Houston, Houston, TX 77030, USA

3 Department of Neurology, Maxine Mesinger Multiple Sclerosis Comprehensive Care Center, Baylor College of Medicine, Houston, TX 77030, USA

* Correspondence: rajesh.k.gupta@uth.tmc.edu; Tel.: +1-832-325-6534

**Abstract:** Erdheim-Chester disease (ECD) is a rare, sporadic, non-Langerhans cell histiocytosis, a multisystem disorder, which has higher mortality when presented with CNS involvement. We report a 46-year-old woman who has ECD with exclusive CNS involvement. She presented with intracranial hemorrhage and had a poor response to corticosteroid and interferon. She required multiple debulking procedures and eventually responded well to cobimetinib. She has not had any other organ involvement thus far. This report highlights that CNS involvement may be the only manifestation of ECD and sometimes may require a repeat biopsy with IHC testing for excellent treatment outcomes.

**Keywords:** Erdheim-Chester disease (ECD); CNS; histiocytosis

## 1. Introduction

Erdheim Chester Disease (ECD) is a rare, non-inherited, non-Langerhans cell histiocytosis characterized by excessive production and accumulation of histiocytes in multiple organs including long bones, the central nervous system, and the kidneys [1]. The pathophysiology of ECD involves mutations in mitogen-activated protein kinase pathways which promote uninhibited histiocyte proliferation. It can have a myriad of clinical presentations, and thus poses a great diagnostic challenge. Diagnosis often involves use of imaging, analysis of histiocytes in tissue biopsies and testing for presence of mutation of the BRAF gene [2]. Treatment may include surgical resection of the lesion, administration of interferon alpha, and targeted therapies such as vemurafenib in BRAF positive and cobimetinib in BRAF negative patients [2].

Here, we report a patient presenting with a subdural hematoma and an intracranial mass requiring serial debulking, who was eventually diagnosed with ECD seven years after her initial presentation. This patient does not have any evidence of extracranial disease activity yet.

## 2. Case Report

A woman in her 40s with hypertension and diabetes mellitus presented with acute onset of right foot drop and headache (HA) in 2013 that she described as the "worst headache of her life". Her physical exam was significant for moderate to severe weakness of the right lower limb. A CT scan of the head showed a 2 cm subdural hematoma at the left anterior falx with a regional mass effect. Intraoperatively, a mass at the falx was found and was biopsied. Postoperative magnetic resonance imaging (MRI) showed "left parafalcine enhancing thickening," which was followed by serial MRIs (Figure 1A–F).

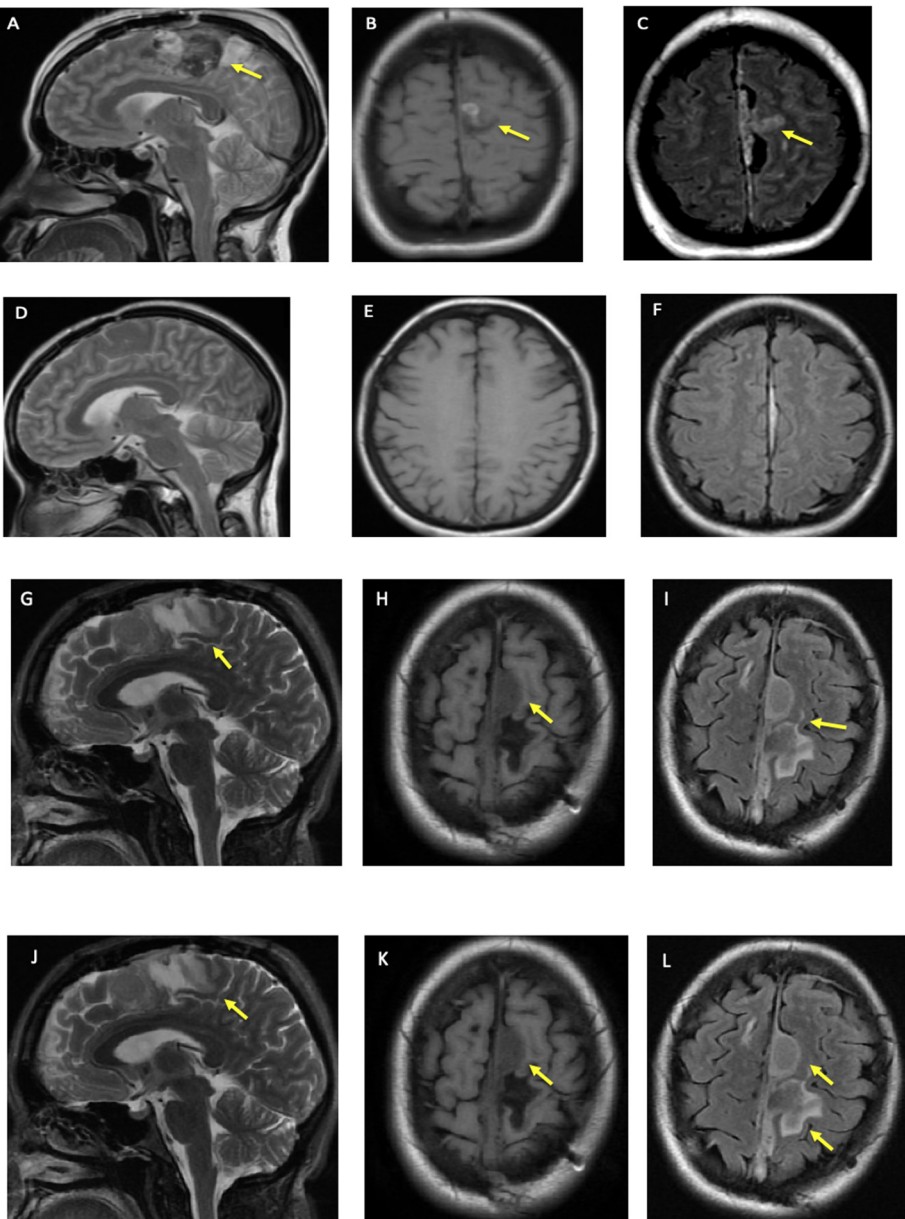

**Figure 1.** Patient's Brain MRI. (**A–C**) show brain MRI at the time of initial presentation in 2013, (**A**) is T2, (**B**) is T1 FLAIR, and (**C**) is T2 FLAIR. (**D–F**) is brain MRI 3 months after the patient's initial craniotomy. (**D**) is T2, (**E**) is T1 FLAIR, and (**F**) is T2 FLAIR. (**G–I**) is 7 years after initial MRI, (after the patient was on cobimetinib for 6 months). (**G**) is T2, (**H**) is T1 FLAIR, and (**I**) is T2 FLAIR. (**J–L**) are 8 years after initial MRI, (on cobimetinib for 13 months). (**J**) is T2, (**K**) is T1 FLAIR and (**L**) is T2 FLAIR. Single yellow arrows in different figures shows the lesion after biopsy and debulking. In (**L**), anterior and first arrow is highlighting meningioma in addition to lesion as denoted by 2nd and posterior arrow.

The gross histology showed a "tan-brown irregular piece of tissue", and microscopic examination revealed a mixed cellular population of mature plasma cells, lymphocytes, and collections of cells with large vesicular nuclei containing prominent nucleoli. The immunohistochemical (IHC) features raised the possibility of ECD.

Three years later, the patient presented with headache and worsening leg weakness. A head CT scan showed a hemorrhage extending along the superior sagittal sinus and left anterior falx with hyperdense left parafalcine mass and mild effacement of the left ventricle. The mass was debulked and biopsied. The IHC panel of this tissue was positive for Factor

XIIIa, suggesting ECD, however negative for BRAF. The whole-body PET CT did not show evidence of involvement of other body systems. The patient was started on pegylated (peg) interferon leading to symptom improvement.

Nearly six years later, she had a recurrence of headache and worsening right leg weakness. An MRI showed an increase in parafalcine mass size and chronic thrombosis of the superior sagittal sinus. The repeat immunohistochemistry showed areas of monocyte infiltration consistent with macrophages and monocytes as shown in Figure 2.

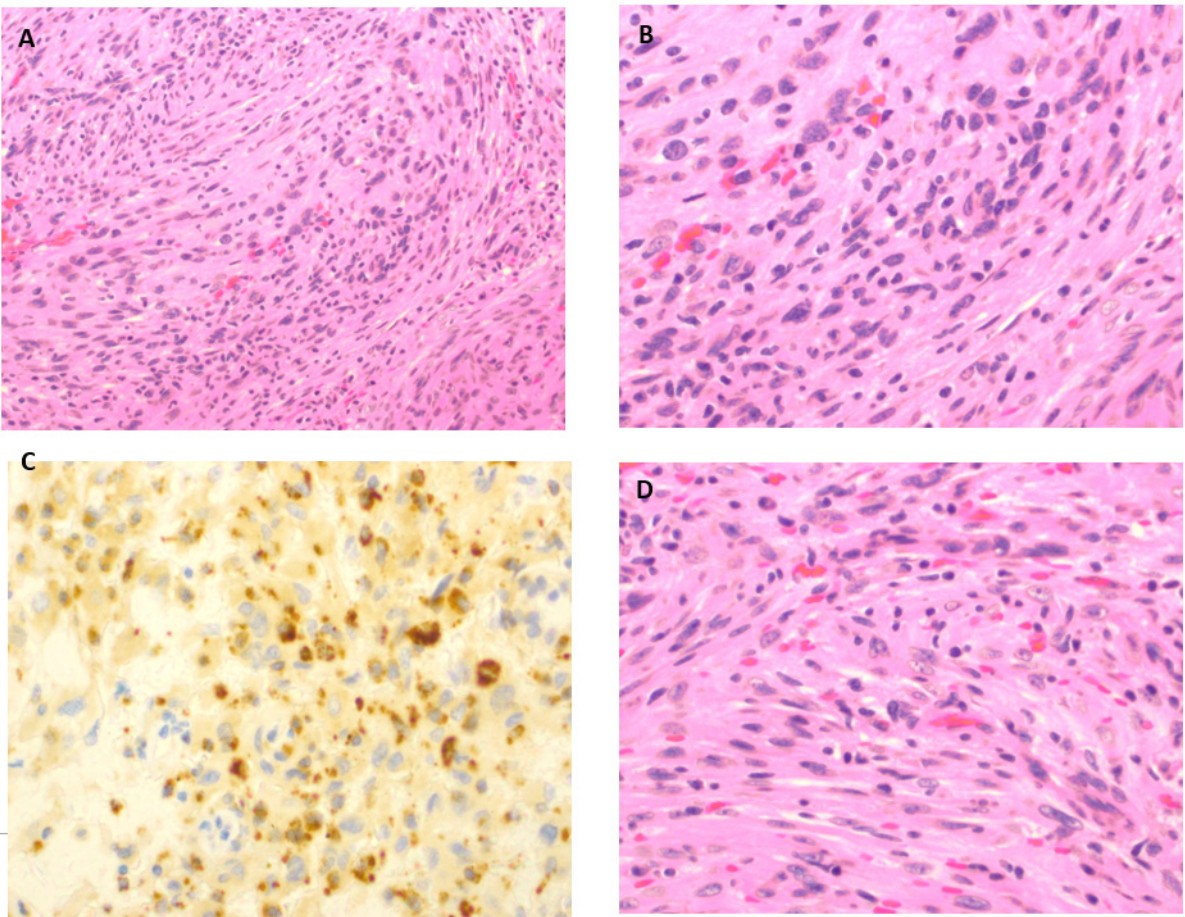

**Figure 2.** Immunohistochemistry pictures from biopsy in 2019. (**A**): Hematoxylin and Eosin 200×, areas of histiocytic and monocyte infiltration. (**B**): Hematoxylin and Eosin 400×, areas of monocyte infiltration consistent with macrophages and monocytes. (**C**): CD68 Immunohistochemistry 400×, Areas of monocyte infiltration consistent with macrophages and monocytes. (**D**): H and E stain 400×, areas of monocyte infiltration consistent with macrophages and monocytes.

The patient had a second debulking procedure and histology revealed "well-formed meningothelial whorls", which was not seen on earlier histopathology. Within a few months, the patient also experienced a focal motor seizure that has remained well controlled on levetiracetam.

In 2020, the patient started cobimetinib, and has not had any progression in clinical and imaging findings (Figure 1G–L). In addition, there are no clinical or radiographic findings to suggest other system involvement over the last eight years.

## 3. Discussion

This patient has several atypical features including an initial presentation with atraumatic subdural hematoma, concomitant meningioma on the last biopsy, poor response to interferon, need for serial debulking, excellent response to cobimetinib, and lack of any

other organ involvement over the last almost eight years. Atraumatic subdural hematoma in the setting of ECD has been reported in the literature before by Noh et al. They described a patient who had multisystem ECD and presented with a subdural hematoma almost a year after ECD diagnosis [3].

Our patient has not had any bone pain or other systemic symptoms. The absence of osseous involvement is supported by the lack of increased tracer uptake during a FDG-PET scan of the axial or extra-axial skeleton. Her most recent histopathology showed meningothelial activity in the form of well-formed meningothelial whorls in the pre-existing background of ECD cells suggestive of the presence of meningioma in addition to the ECD mass. Although dural involvement is common in ECD, concomitant meningioma has not been reported in the literature before. Additionally, our patient did not have other common neurological symptoms of cognitive impairment, peripheral neuropathy, or cerebellar ataxia.

Our patient did not respond to treatment with interferon and continued to have disease progression similar to findings of some studies in the past, which reported continued disease progression after treatment with corticosteroids and interferon [4–7].

She was not a candidate for Vemurafenib as she was BRAF negative. She has not shown any progression of symptoms or lesions on imaging since she started cobimetinib in 2020. Cobimetinib is an allosteric inhibitor of MEK1 and MEK2 proteins and was FDA-approved for BRAF-negative ECD patients in 2019. A 2019 study by Diamond et al. showed 89% of patients with histiocytic neoplasms treated with cobimetinib to have complete resolution of the disease or stabilization of the disease symptoms, with 72% of patients experiencing complete metabolic response (as assessed by FDG-PET scan) [8]. Cobimetinib is associated with toxicities, including photosensitivity, acneiform rash, and rhabdomyolysis [9].

The fact that our patient is BRAF negative may explain why she did not show a response to interferon but did to cobimetinib, which is a particularly effective therapy for BRAF negative ECD variants. Similar to our patient, poor response to interferon has been noted in ECD patients with predominant CNS or cardiac involvement by Haroche et al. [10].

In addition to our patient, we found three patients in the literature who had ECD limited to the CNS. One patient, reported by Pan et al., had a BRAF-positive parenchymal mass and responded to vemurafenib treatment [11]. Two patients, published by Wagner et al., had BRAF negative variants—patient 1 improved after debulking surgery, and patient 2 experienced disease progression after treatment with interferon and lesion debulking [12]. Our patient is distinct from the patient described by Pan et al. as she was not eligible for vemurafenib therapy. Furthermore, she is distinct from Wagner et al.'s patient 1 in that she did not have disease resolution with debulking, and in contrast to patient 2, she received additional cobimetinib therapy, that at the time of Wagner et al.'s publication, was not yet FDA approved for ECD therapy. Compared to patient 2 described by Wagner et al., our patient has experienced disease stabilization after cobimetinib, whereas patient 2 continued to experience disease progression after interferon therapy and debulking surgery.

This case report highlights the importance of considering ECD in differential diagnoses when a patient presents with CNS symptoms of unclear etiology. Additionally, this emphasizes the importance of biopsy and IHC testing of cellular pathology to determine the correct course of treatment. Furthermore, this paper reinforces data found in Diamond et al.'s study that cobimetinib effectively prevents ECD disease progression.

**Author Contributions:** This manuscript was approved by all authors and represents valid work; neither this manuscript nor one with substantially similar content under our authorship has been published or is being considered for publication elsewhere. We certify that we are the sole authors of this paper and hereby take public responsibility for the entire content of the manuscript. Conceptualization R.K.G. and A.H. methodology, R.K.G.; software, A.H. and R.K.G.; validation, A.H. and R.K.G.; formal analysis, A.H. and R.K.G.; investigation, A.H. and R.K.G.; resources, A.H. and R.K.G.; data curation, A.H., R.K.G., T.A.R. and R.K.G.; writing—original draft preparation, A.H., C.A.P. and R.K.G.; writing—review and editing, all authors; visualization, all authors.; supervision, R.K.G.; project administration, A.H. and R.K.G.; funding acquisition, no funding. All authors have read and agreed to the published version of the manuscript.

**Funding:** This research received no external funding.

**Institutional Review Board Statement:** Because this case report does not contain protected health information, ethical approval was not obtained.

**Informed Consent Statement:** Although no identifying patient information is used, an informed consent was obtained from the patient referred to in this article.

**Data Availability Statement:** Not applicable.

**Acknowledgments:** We would like to express our gratitude to Christie Lincoln and Jerry Goodman for their help with the review of the patient's MRI and providing histopathology images, respectively. We also would like to thank Ishita Agarwal DDS for editing and proof reading of this article.

**Conflicts of Interest:** The authors declare no conflict of interest.

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
