# Peer review of "A Patient with Erdheim-Chester Disease Limited to Central Nervous System"

_2035-8377, doi:10.3390/neurolint14030056_

Round 1

Reviewer 1 Report (Previous Reviewer 2)

Both are fine

Author Response

We greatly appreciate your review.  Authors have reviewed manuscript and edited as needed. Thanks !

Reviewer 2 Report (New Reviewer)

Authors present a case of Erdheim-Chester disease with manifestations limited to Central Nervous System, which is a rare finding, since the disease shows a spread involvement of multiple systems, with the skeletal above all. The results presented summarize the findings of several years of follow-up. I have few main and minor concerns:  

- On line 38-39 there is mention of the lesion on surgical exploration, but no MRI preliminary finding; was it performed?

- On line 91 authors mention "no peripheral neuropathy", but no electroneurographic finding is mentioned; 

- introduction must be widened a little; 

- there is no mention of any preventive or acute treatment for headache control; 

As a minor concern, authors may consider citing the following recent review "Haroche J, Cohen-Aubart F, Amoura Z. Erdheim-Chester disease. Blood. 2020 Apr 16;135(16):1311-1318. doi: 10.1182/blood.2019002766. PMID: 32107533".

Author Response

We greatly appreciate reviewer's comments and suggestions. Here are our responses.

-On line 38-39 there is mention of the lesion on surgical exploration, but no MRI preliminary finding; was it performed? 

response:  thanks; No, MRI was not performed before surgery given urgent need for surgery.

-On line 91 authors mention "no peripheral neuropathy", but no electroneurographic finding is mentioned;

response:  we meant "no symptoms of peripheral neuropathy"; No NCS or EMG was performed performed.

introduction must be widened a little;

response  Great idea. We have elaborated the introduction section.

- there is no mention of any preventive or acute treatment for headache control; 

response: thanks, headache was mostly in acute setting and improved with debulking and treatment of intracranial lesion.

As a minor concern, authors may consider citing the following recent review. "Haroche J, Cohen-Aubart F, Amoura Z. Erdheim-Chester disease. Blood. 2020 Apr 16;135(16):1311-1318. doi: 10.1182/blood.2019002766. PMID: 32107533".

response:  Thanks for great suggestion. we have cited this now as no. 2 reference.

Round 2

Reviewer 2 Report (New Reviewer)

Authors presented an interesting case of Erdheim-Chester Disease limited to Central Nervous System, very interesting and with a very good quality of presentation. At this time I have no concern about the paper. 

This manuscript is a resubmission of an earlier submission. The following is a list of the peer review reports and author responses from that submission.

Round 1

Reviewer 1 Report

Dear authors,

I would like to thank you for giving me the opportunity to review a manuscript for your journal, and I hope that I provided concise and meaningful remarks. It was a pleasure to read and review the manuscript “Erdheim-Chester Disease with Isolated CNS Involvement: A Case Report and Systematic Review of Literature"

My comments are the following:

1)I would suggest that authors should add in table 4 the paper by Kaiafa G. et al. Erdheim-Chester Disease during the COVID-19 Pandemic. Medicina (Kaunas). 2021 Sep 22;57(10):1001 with a BRAF positive ECD with retroperitoneal localization.

2)You should highlight that in many cases repeated biopsies are required to conclude in the final diagnosis

3) You should further empower their diagnosis since your case was BRAF negative but also the latest biopsy demonstrated meningioma evidence (2 different histopathological diagnosis?)

Author Response

Comments to Reviewer 1

1)I would suggest that authors should add in table 4 the paper by Kaiafa G. et al. Erdheim-Chester Disease during the COVID-19 Pandemic. Medicina (Kaunas). 2021 Sep 22;57(10):1001 with a BRAF positive ECD with retroperitoneal localization.

Response: We greatly appreciate the time you took to review this paper and your thoughtful comments.

The patient mentioned in the above paper (Kaiafa G. et al) did not have CNS involvement and we believe that it will not be a good fit for this table 4, which provides documentation of the frequency of neurological symptoms in patients with ECD who presented with CNS symptoms. Besides that this table includes case series and other studies that have data of multiple patients.

 We also thought about adding this paper to the discussion section but this didn’t look appropriate. We will be happy to reevaluate our decision as to the reviewer's wish.

2) You should highlight that in many cases repeated biopsies are required to conclude in the final diagnosis

Response: Appreciate this great suggestion. Yes, we have highlighted this.

3) You should further empower their diagnosis since your case was BRAF negative but also the latest biopsy demonstrated meningioma evidence (2 different histopathological diagnosis?)

Response: Thanks for noticing this and letting us know. The final and 3rd biopsy actually showed findings of both histiocytosis as well as superimposed meningioma. We have clarified this in the revised manuscript.

Reviewer 2 Report

This systematic literature review of isolated cerebral manifestation of ECD is timely and provides important insights to this rare and challenging condition. The methods are sound and the case reports adds an important component to this manuscript. I just have a few but compulsory remarks:

  • The headings of tables/figures need to be rephrased
  • Means and SD need to be added to the tables where appropriate
  • Figure 5: „torso“ is not the appropriate wording
  • The current Figure 5 can be used for any neurological condition, which is inappropriate. Specify the condition in which this work-up is suggested. If I understand correctly, you would suggest to make the diagnosis of isolated cerebral ECD without histology, which is also inappropriate. The entire figure needs tob e restructured.

Author Response

Comment to Reviewer 2

  • Comment: The headings of tables/figures need to be rephrased
  • Response: We greatly appreciate the time you took to review this paper and your thoughtful comments. We made the recommended changes to the wording of the captions of the figures and tables. 

  • Comment: Means and SD need to be added to the tables where appropriate
  • Response: Thanks for letting us know. We have added standard deviation where appropriate.

  • Comment: Figure 5: „torso“ is not the appropriate wording.
  • Response: Appreciate you noticing this. We have corrected this to CT chest, abdomen and pelvis

  • Comment: The current Figure 5 can be used for any neurological condition, which is inappropriate. Specify the condition in which this work-up is suggested. If I understand correctly, you would suggest to make the diagnosis of isolated cerebral ECD without histology, which is also inappropriate. The entire figure needs to be restructured.

  • Response: Greatly appreciate your suggestion. Figure 5 has also been re-worded and restructured. We are happy to make any further changes if needed.

Round 2

Reviewer 2 Report

no further comments